# Automatic methods of hoof-on and -off detection in horses using wearable inertial sensors during walk and trot on asphalt, sand and grass

Eloise V. Briggs[1,2]*, Claudia Mazzà[1,2]

1 Department of Mechanical Engineering, University of Sheffield, Sheffield, United Kingdom, 2 INSIGNEO Institute for in silico Medicine, University of Sheffield, Sheffield, United Kingdom

* evbriggs1@sheffield.ac.uk

**Data Availability Statement:** The data underlying this study are available on Figshare (10.15131/ shef.data.14945844).

## Abstract

Detection of hoof-on and -off events are essential to gait classification in horses. Wearable sensors have been endorsed as a convenient alternative to the traditional force plate-based method. The aim of this study was to propose and validate inertial sensor-based methods of gait event detection, reviewing different sensor locations and their performance on different gaits and exercise surfaces. Eleven horses of various breeds and ages were recruited to wear inertial sensors attached to the hooves, pasterns and cannons. Gait events detected by pastern and cannon methods were compared to the reference, hoof-detected events. Walk and trot strides were recorded on asphalt, grass and sand. Pastern-based methods were found to be the most accurate and precise for detecting gait events, incurring mean errors of between 1 and 6ms, depending on the limb and gait, on asphalt. These methods incurred consistent errors when used to measure stance durations on all surfaces, with mean errors of 0.1 to 1.16% of a stride cycle. In conclusion, the methods developed and validated here will enable future studies to reliably detect equine gait events using inertial sensors, under a wide variety of field conditions.

## 1 Introduction

In both equine sport and medicine, there is increasing demand for quantitative analysis of gait under field conditions [1]. Many techniques are predicated on the reliable detection of gait cycle events: *hoof-on* and *-off* [2]. These *gait events* are the instants in a gait cycle when the hoof first comes into contact with the ground (hoof-on) and that when it is first lifted fully from the ground (hoof-off) [3].

Methods to accurately detect gait events are valuable in various applications including performance analysis and lameness quantification. For instance, an increase in positive diagonal advanced placement (in which the hoof-on of the hindlimb precedes that of the contralateral forelimb) has been found to be an indicator of superior gait quality in advanced dressage horses [4] and approved Warmblood stallions [5]. Furthermore, the timing of gait events can be used to calculate the suspension of flying gaits, in which all four hooves are off the ground simultaneously.

**Funding:** The research of EVB is funded by Worldbase Ltd. and this study was also supported by the UK EPSRC (EP/K03877X/1, EP/S032940/1, https://epsrc.ukri.org). The funders had no role in study design, data collection and analysis, decision to publish, or preparation of the manuscript.

**Competing interests:** The research of EVB is funded by Worldbase Ltd. and this study was also supported by the UK EPSRC (EP/K03877X/1, EP/S032940/1, https://epsrc.ukri.org). The funders had no role in study design, data collection and analysis, decision to publish, or preparation of the manuscript. The commercial funder Worldbase Ltd. is a manufacturing company, specialising in agricultural machinery and currently has no products or services, existing or in design, related to the content of this research. This does not alter our adherence to PLOS ONE policies on sharing data and materials.

Studies into lameness detection have consistently found there to be a link between a statistically significant reduction in suspension phase and the presence of lameness [6–8].

The gold standard of gait event detection remains the force plate [9], which can directly measure ground reactions to differentiate between the load-bearing stance phase of a limb and swing phase. However, force plates not only come with extremely high costs but limit the number of strides which can be analysed, requiring a single foot to be placed fully on the plate for reliable analysis. Instrumented treadmills [10] have been used to overcome the latter point but these are often unsuitable for simulating real-world scenarios. Although a very limited number of studies have now used force plates in field conditions [11,12], the obvious complexities of integrating them into surfaces limit wider adoption of such methods. These factors make force plates unsuitable for collecting data under field conditions.

As such, efforts have been made to develop methods which use portable devices. Particularly, wearable inertial measurement units (IMUs) have been heralded as a potential solution to the problem of gait event detection in the field, being relatively inexpensive and highly convenient, (compared to the alternative force plate or optical motion capture systems). They enable data collection over any range of distance and conditions, are comparatively easy to attach to subjects, reducing set-up times, and are less cumbersome than some alternatives, minimising effect on the horse's movement.

Tijssen et al. [13] reported that IMUs attached to the lateral hoof wall were capable of successfully identifying gait events, using linear accelerations and angular velocities. Distinctive peaks in the resultants of acceleration and angular velocity were found to coincide with instances of hoof-on and -off recorded simultaneously by a force plate. Whilst this method might offer the most accurate and precise detection of gait events, securing sensors to the hooves is not always practicable- attachment being time consuming and there being a high risk of damage to sensors during data collection. Conversely, most horses very quickly become acclimatised to boots and wraps worn on the cannons or pasterns; hence, this paper sought to develop methods of gait event detection which used sensors mounted in these locations. Several previous studies have used inertial devices attached at the level of the pasterns to identify strides [14–16], and there is a commercial system which uses these methods (Lameness Locator™, Equinosis, Columbia). However, to the authors' best knowledge, there are no published methods for specific gait event detection which are publicly available.

Current state of the art methods use IMUs attached at the level of the cannon bone [9] or upper body locations [17] but validation of these have been limited in terms of subject cohorts and surface conditions. Different breeds of horse have inherently different conformation [18] which can have a significant effect on gait kinematics [19]. Age can also affect kinematics, with older horses showing signs of stiffened gait, for example due to osteoarthritis [20]. Whether the horse is shod or not can also affect gait [21]. Furthermore, extensive research has found that surface type has a significant effect on the horse's kinematics [22]. Despite these reported variations, previous studies have commonly included only one breed of horse and considered only one hard, concrete-like flooring.

The aim of this research was to propose and validate IMU based methods of gait event detection, including different sensor locations and different exercise surfaces. This will facilitate future studies under a variety of field conditions.

## 2 Materials and methods

### 2.1 Subjects

Eleven horses (eight geldings and three mares; of mean and standard deviation (SD) heights and ages of 154(21)cm and 12(8)years, respectively) of various breeds and levels of training

**Table 1. Details of subject population.**

| Horse | Breed/type | Height (cm) | Age (years) | Shod (y/n) | Usual use |
|---|---|---|---|---|---|
| 1 | Selle français | 158 | 11 | y | Low level eventing |
| 2 | ISH | 166 | 13 | y | 5* eventing |
| 3 | ISH | 147 | 20 | y | Low level eventing |
| 4 | Friesian | 158 | 4 | y | Hacking |
| 5 | Westfalian | 168 | 7 | y | Low level eventing |
| 6 | ISH | 168 | 14 | y | 2* eventing |
| 7 | Shetland | 91 | 30 | n | Retired |
| 8 | ISH | 168 | 6 | y | Low level eventing |
| 9 | Cob | 150 | 12 | n | Hacking |
| 10 | Welsh C | 149 | 16 | n | Low level dressage |
| 11 | ISH | 166 | 5 | y | Low level eventing |
| Mean(SD) | - | 154(21) | 12(8) | - | - |

Details include whether each horse was shod or not (reported y or n, respectively) and the usual use of the horse; ISH indicates Irish Sports Horse.

and fitness, from retired ponies to top level event horses, were included in the study (Table 1). Eight horses were shod and three were not.

## 2.2 Data acquisition

Six IMUs (Shimmer3 IMU, Shimmer Sensing, Dublin) were used to record data at 200Hz. Sensors were attached laterally to the hooves and the regions of the pasterns (proximal phalanges) and cannon bones (third meta-carpal and -tarsal bones) (Fig 1) of the left fore- and hindlimbs. Gyroscopes and magnetometers recorded data ranges of ±2000deg/s and ±49Ga, respectively, and the triaxial accelerometers of ±16g at the cannons and ±200g at the pasterns and hooves. Cannon sensors were attached using commercially available pockets (Estride™, Bognor Regis), pastern IMUs using Velcro wraps and tape, and hoof IMUs using tape and elasticated bandages.

The horse was led by an experienced handler in a straight line of 25m at walk and trot, three passes per gait, on a hard control surface (asphalt). Trials were repeated on a grass field (grass) and sand and rubber chip surface (sand). Efforts were made to ensure data was

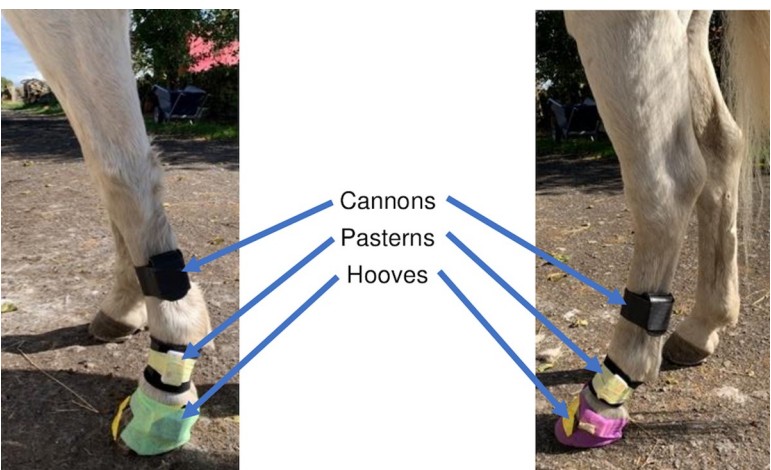

**Fig 1. IMU placement.** Sensors were attached to the cannons, pasterns and hooves of the left fore- and hindlimbs using custom-made pockets, Velcro wraps and elasticated bandages, respectively, and secured with tape.

recorded under similar conditions for all horses. All the surfaces were level and datasets were collected during periods of dry weather when the grass field would be firm and the sand soft. Horses were verbally encouraged to maintain the correct gait (walk or trot) but were allowed to move at self-selected speeds. The methods were reviewed and approved by The University of Sheffield, Ethics Department (Reference Number 033398), and horse owners gave signed consent for their animal's involvement.

### 2.3 Gait events detection

The gait events (Fig 2) were defined as the instants in the stride cycles when the hoof first comes into contact with the ground at the onset of the stance phase (hoof-on) and when it is lifted from the ground, following break-over (hoof-off) [3]. A stride cycle refers to one full cycle of gait, from one hoof-on to the subsequent hoof-on.

In the data processing stages, the timings of gait events were estimated using various processing methods and data from different sensor locations. Algorithms were developed in MATLAB (version 2020R, The MathWorks Inc., Natick, Massachusetts, USA).

**2.3.1 Reference method.** Data from hoof mounted IMUs were processed as per Tijssen et al. [13] and gait event timings determined using this method ($M_{ref}$) were used as the reference values against which to compare those obtained using other methods. $M_{ref}$ was previously robustly validated against a gold standard, (i.e. a force plate) for fore- and hindlimb gait events at walk and trot. Accuracies in the range of 2.4ms to 12.2ms for hoof-on and 3.2ms for hoof-off were reported [13].

Briefly, $M_{ref}$ assumes that a prominent peak in the resultant of angular velocity ($AngVel_R$) measured at the hoof signifies a hoof-on (Fig 3C, down triangles). A prominent peak in the resultant acceleration ($Acc_R$) arises at instances of hoof-off (Fig 3F, up triangles).

**2.3.2 Alternative methods.** The first novel method estimated hoof-on and -off using $Acc_R$ of either the pastern ($M1_p$) or cannon ($M1_c$). At the level of the hoof, peaks in $Acc_R$ are created by the hoof-surface impact (hoof-on) and subsequent hoof lift-off (hoof-off) [13]. Here, it is hypothesised that these peaks in acceleration, which correspond to the gait events, would be detectable at the more proximal locations of the pasterns and cannons.

Both $M1_p$ and $M1_c$ were applied for fore- and hindlimbs at walk and trot. To assist in the subsequent peak detection, $Acc_R$ was first segmented into rough periods of midstance to midstance (Fig 3A and 3D, blue windows). To achieve this for walk, a 1D median filter [23] with

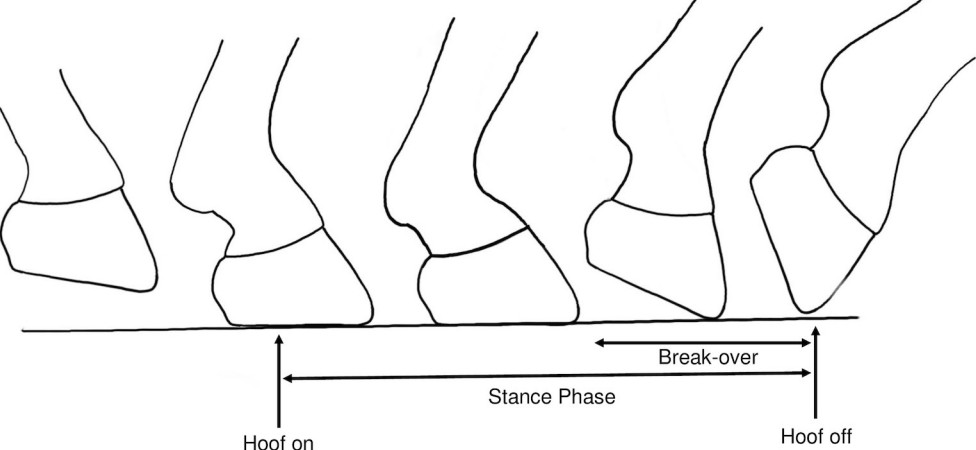

**Fig 2. Illustration of the gait events.** Position of the hoof at the instants of hoof-on and -off and how these gait events relate to the stance phase and break-over of the gait cycle.

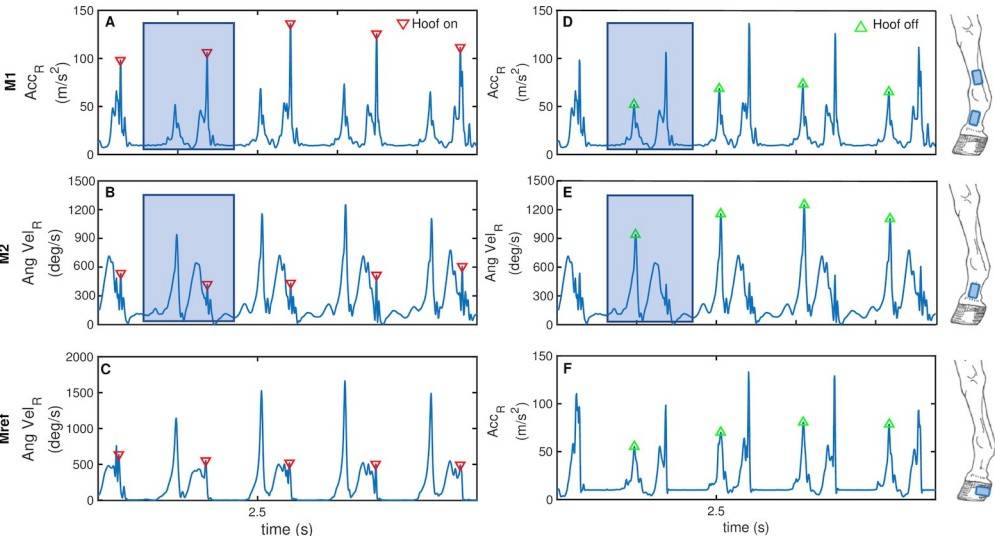

**Fig 3. Illustration of the novel methods.** Novel methods M1 (A and D) and M2 (B and E), and reference method $M_{ref}$ (C and F) are illustrated; blue windows are examples of the frames in which relevant peaks were sought; down triangles (A-C) indicate hoof-on and up triangles (D-F) hoof-off; the sensor locations (hooves, pasterns and/or cannons) for which each method is applicable are shown in illustrations on the right.

window length of half the sampling frequency followed by a 2nd order Butterworth filter with cut off frequency of 5Hz was used to identify rough locations of the swing and stance periods. For trot, the same median filter followed by a 2nd order Butterworth filter with 20Hz cut off frequency was applied to the hindlimb cannon data, whereas a Butterworth filter with 5Hz cut off frequency, alone, was used for all other trot cases. The raw $Acc_R$ was then filtered with a 2nd order Butterworth filter, cut off frequency 40Hz and one prominent peak near the beginning (Fig 3D, up triangles) and one near the end of each window (Fig 3A, down triangles) were labelled as hoof-off and hoof-on, respectively.

The second novel method estimated hoof-on and -off using $AngVel_R$ recorded at the pastern ($M2_p$) mounted sensors. Tijssen et al. [13] reported that spikes in $AngVel_R$ recorded at the hoof coincide with hoof-on and -off. Here, the hoof wall can be considered a rigid structure meaning there would be no angular movement of the hoof relative to the solid ground during stance phases. Hence, peaks in $AngVel_R$ before and after the stance duration, where the signal is quite flat, coincide with hoof-on and -off. On this premise, it was hypothesised that a peak corresponding to the gait events would also be detectable in $AngVel_R$ at the level of the pasterns.

$M2_p$ was applied for the fore- and hindlimbs at walk and trot. After applying the same windowing method described for M1 to the $AngVel_R$ signal, but with median filter window length of quarter the sampling frequency (Fig 3B and 3E, blue windows), the raw $AngVel_R$ was filtered using a second order Butterworth filter, cut off frequency 40Hz. A prominent peak near the beginning (hoof-off, Fig 3E, up triangles) and one near the end (hoof-on, Fig 3B, down triangles) of each window were detected.

Two additional methods, taken from the literature, were also investigated. Method $M3_c$ estimates hoof-on and -off using the angular velocity measured at the cannon [24]. The raw component of angular velocity about the sensor axis aligned with the mediolateral direction of the horse ($AngVel_{ML}$) recorded from a cannon mounted sensor is used. In this method, previously validated against a motion capture system for forelimb gait events at trot (Fig 4B), the deepest trough following the large peak of the signal is used to identify the hoof-on (down triangles) and the first peak to occur after the relatively flat (stance) portion to identify the

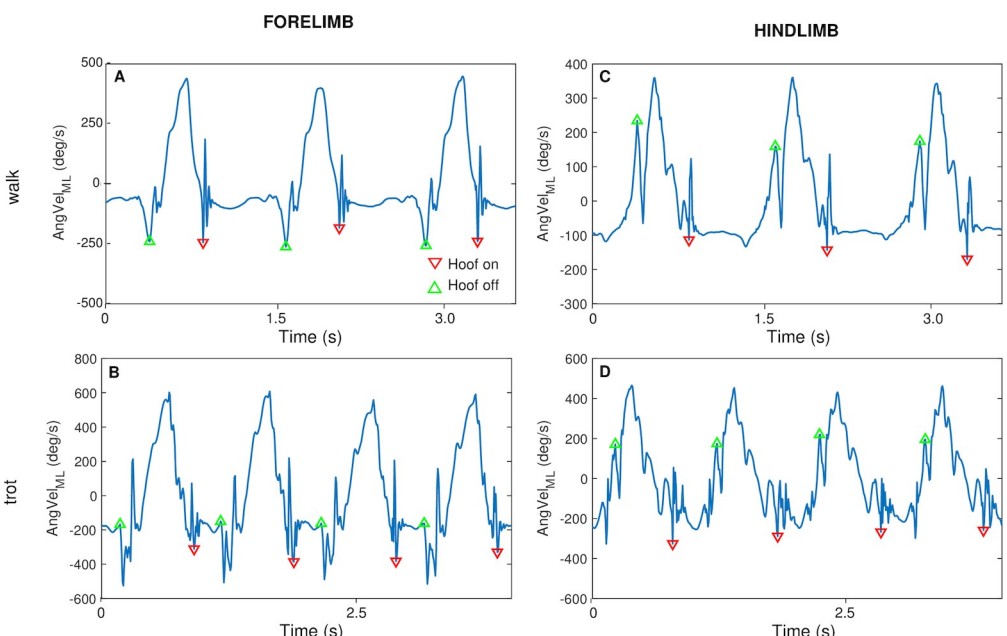

**Fig 4. Illustration of M3$_c$.** The implementation of M3$_c$ for each limb and gait is illustrated; hoof-on events are indicated by down triangles and hoof-off by up triangles.

hoof-off (up triangles). In this paper, the method was also applied to the data from the hindlimbs and walk, with adjustments made owing to the different AngVel$_{ML}$ signal profiles produced by the different limb and gaits. For each limb and gait, the biggest peak in the signal was observed during the swing. For the hind limbs at trot (Fig 4D), the deepest trough following this large peak was again detected as the moment of hoof-on and the last peak before it was used as the hoof-off. For the forelimb at walk (Fig 4A), the deepest troughs before and after the large peak, respectively, were detected as instances of hoof-off and on. For the hindlimbs at walk (Fig 4C), the deepest trough after the large peak was taken as the time of hoof-on and the last peak before it as the time of hoof-off.

Method M4$_c$, taken from [9], was applied to estimate hoof-on and -off from orientation and acceleration cannon data at walk and trot for fore- and hindlimbs, for which it was previously validated against force plate methods. First, the sensor angles were calculated using the IMUs' proprietary software (ConsensysPRO, Shimmer Sensing, Dublin). The timings of midswing and -stance points were estimated form these angles and these were then used to assist in detection of peaks corresponding to hoof-on and -off events in the Acc$_R$.

## 2.4 Data analysis

**2.4.1 Quantification of errors in event detection.** Descriptive statistics and agreement were calculated in Matlab. The initial analysis was conducted only on events recorded on the control surface- asphalt, considered the reference surface. Errors (E$_{on}$ and E$_{off}$) were calculated (Eqs 2.1 and 2.2) by comparing hoof-on and -off events detected by the reference method (Hon$_{ref}$ and Hoff$_{ref}$) to those detected by the alternative method (Hon$_{alt}$ and Hoff$_{alt}$).

$$E_{on} = Hon_{ref} - Hon_{alt} \tag{2.1}$$

$$E_{off} = Hoff_{ref} - Hoff_{alt} \tag{2.2}$$

For individual gait events, the errors were expressed in ms. The performance of each method was assessed in terms of both accuracy and precision. Accuracy was defined as the means of the errors, $E_{on}$ and $E_{off}$, incurred by the method and precision as the standard deviation of these errors. Methods were considered superior if the values of accuracy and precision were low, indicating a low mean error and small distribution of the error.

Gait events detected by each method were used to calculate stride durations. The agreement of each with the reference method was quantified by the limits of agreement (LoA), with upper and lower limits of agreement (ULoA and LLoA) calculated as per Bland and Altman [25] (Eqs 2.3 and 2.4), and the intraclass correlation coefficient for interrater reliability, ICC{3,1} [26]. Incurrences of false positive (FP), false negative (FN) and true positive (TP) events were detected and used to calculate sensitivity (Eq 2.5) and positive predictive value (PPV) (Eq 2.6) [27].

$$ULoA = E_{mean} + 1.96 \cdot SD \tag{2.3}$$

$$LLoA = E_{mean} - 1.96 \cdot SD \tag{2.4}$$

$$sensitivity = \frac{TP}{TP + FN} \cdot 100 \tag{2.5}$$

$$PPV = \frac{TP}{TP + FP} \cdot 100 \tag{2.6}$$

Statistical analysis was carried out in IBM SPSS Statistics V27.0 (Armonk, NY), with p-values <0.01 indicating significance. Differences between mean errors ($E_{mean}$) for $M1_p$, $M2_p$ and $M1_c$ were tested for significance. Normality of the error data was evaluated using Shapiro-Wilks test and by checking the Q-Q plots. Where normality was upheld, a one-way repeated measures ANOVA, controlled for the covariate *individual horse*, was conducted and Bonferroni post hoc test used to identify the source of significance, if any were found. For datasets which violated the assumption of homogeneity of variance, the Games-Howell post hoc test was used. Data which violated the assumption of normality was compared for significance using a Friedman test and Wilcoxon Signed Rank post hoc test if significant differences were identified.

**2.4.2 Comparison between different surfaces.** After the most accurate and precise methods were identified for each gait event, they were used to calculate stance durations (T) for the fore- and hindlimbs at walk and trot on all surfaces (Eq 2.7, where n is the number of the current stride):

$$T = Hoff_n - Hon_n \tag{2.7}$$

Each stance calculated from the novel method ($T_{alt}$) was compared to that obtained by the reference method ($T_{ref}$) by Eq 2.8 to obtain a value of error ($E_{stance}$).

$$E_{stance} = T_{ref} - T_{alt} \tag{2.8}$$

Bland-Altman methods were used to investigate the effect of different surfaces on the error incurred in stance calculation. Errors were expressed as percentages of the total stride duration and analysed as Bland-Altman figures, with $E_{stance}$ plotted against $T_{mean}$ (Eq 2.9).

$$T_{mean} = \frac{T_{ref} + T_{alt}}{2} \tag{2.9}$$

The difference in mean $E_{stance}$ incurred, for each limb and gait, on each of the three surfaces were compared using a repeated measures one-way ANOVA or Friedman test, depending on the result of a Shapiro-Wilks test for normality and consideration of the Q-Q plots. Again, Bonferroni and Wilcoxon signed rank post hoc tests were used in each case. The effect of the individual horse covariate was controlled.

## 3 Results

A total of 1465 walk and 1255 trot strides were analysed. These were 500, 535 and 430 walk strides and 438, 399 and 418 trot strides on asphalt, grass and sand, respectively.

### 3.1 Event detection

The accuracies and precisions (mean and SD, as per previous definition) of the methods to detect gait events (Fig 5) and agreement (LoA and ICC) with the reference method are given in ms (Table 2).

For the cannon data, $M1_c$ performed better than $M3_c$ and $M4_c$, in all cases (Fig 5E–5H and 5M–5P). The values of ICC (Table 2) showed excellent agreement for all uses of $M1_p$, $M2_p$, $M1_c$ and $M3_c$ with $M_{ref}$ to calculate stride durations (ICC>0.90). Agreement for $M4_c$ was not consistently excellent, with the poorest value (ICC<0.55) occurring for hindlimb hoof-on events at trot. The poor agreements seen for $M4_c$ along with the high mean errors and SD, led to exclusion of the method from further analysis.

For the pastern data, $M1_p$ and $M2_p$ both performed well for detecting all gait events (Fig 5A–5D and 5I–5L). $M1_p$ generally outperformed $M2_p$, except in the case of the forelimb events at trot, where $M2_p$ incurred a significantly smaller mean error (p = 0.005, hoof-on, Fig 5C; and p<0.001, hoof-off, Fig 5D). For hindlimb hoof-on events (Fig 5K), the two methods were equally successful with $M1_p$ detecting events slightly late by a mean(SD) delay of 2(10)ms, and $M2_p$ slightly early, -1(10)ms. $M1_p$ was generally more successful than $M1_c$ except in the case of hoof-off events at walk (p = 0.130, forelimb, Fig 5B and 5F; and p<0.001, hindlimb, Fig 5J and 5N). For the novel methods, no significant between-horses effects were found (p>0.9).

Overall, the pasterns appear to be a superior location for sensor attachment than the cannons. In terms of reliability, pastern methods consistently achieved sensitivity and PPV values of 99 or 100%. $M1_p$ was identified as the better method for detecting all gait events except forelimb events at trot, for which $M2_p$ demonstrated better accuracy. These were the methods chosen to detect the gait events for calculation of stance durations on the control surface and the two additional surfaces for the next level of analysis.

### 3.2 Comparison between surfaces

The errors incurred in stance durations for asphalt, sand and grass are summarised in Table 3 as percentages of total stride duration.

Reliability of the pastern methods remained high when applied to the additional surfaces, maintaining sensitivity and PPV values of 97% and above. $M1_p$ performed equally well when applied to the additional surfaces as well as asphalt and small variations in mean were not statistically significant. At trot $M2_p$ performed slightly better on grass and sand compared to the control surface and with smaller SD in both cases. Forelimb stance durations at trot were the least precise except for the hindlimbs at walk on grass.

Results of the Bland-Altman analysis are shown in Fig 6. The mean error, LLoA and ULoA were calculated after pooling together stance durations on all three surfaces. Although the LoA, calculated in this way, were wider than for each of the three surfaces considered individually, the mean error remained close to 0% and LoA only exceeded 10% for the forelimbs at trot

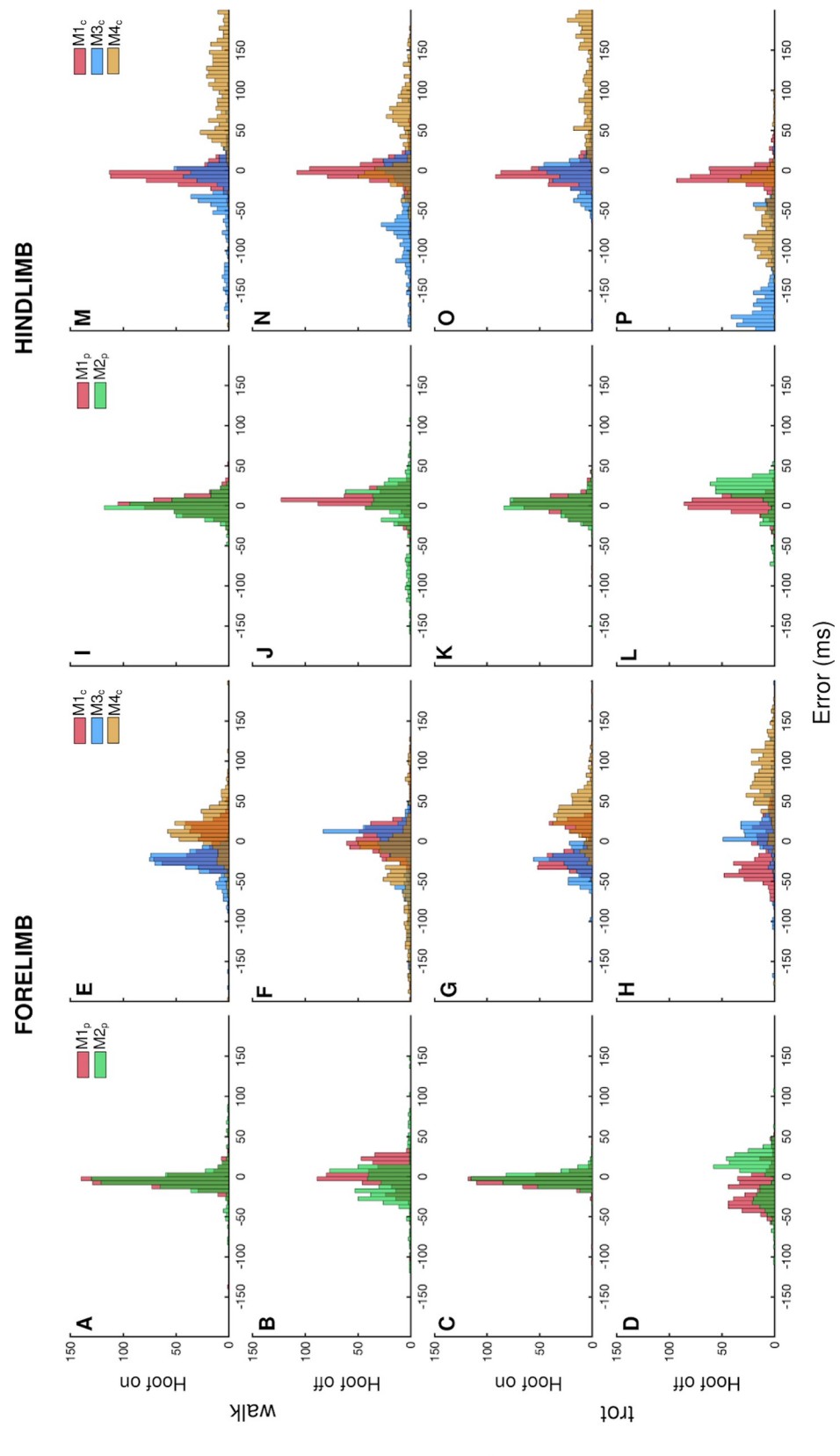

**Fig 5. Distribution of errors incurred by methods to detect gait events.** Histograms of the errors ($E_{on}$ and $E_{off}$) in ms incurred by each method in detecting each of the eight types of gait events.

(Fig 6B). The higher LoA in the latter case are reflective of the higher values of SD (Table 3). As with event detection on asphalt, no significant between-horses effects were seen (p>0.8), suggesting methods had the same performance for all individuals.

## 4 Discussion

The aim of this research was to propose and validate different methods of gait event detection using IMUs, exploring both different sensor attachment locations and signal processing techniques. Overall, the best results were obtained from the pastern sensor using an algorithm based on the analysis of the resultant acceleration ($M1_p$), with the only exception being the forelimbs at trot, where angular velocity ($M2_p$) was shown to be preferable. These methods proved superior to those cannon-based ones previously introduced in the literature ($M3_c$ and $M4_c$). Next, the study proved that the validity of these two methods held on different surface types.

### 4.1 Event detection

Cannon-based methods, as previously proposed in the literature, proved to be accurate but not precise at walk and less accurate at trot, despite having previously been validated under this condition. $M3_c$ errors in detecting forelimb hoof-on and -off events, were -30ms and -15ms, respectively, at walk and -26ms and 62ms at trot. Similar errors were found for the hindlimb hoof-on events at walk (-31ms) and smaller errors for the hindlimb hoof-on at trot (-4ms). However, larger errors were observed for hoof-off events for the hindlimb at both walk (-31ms) and trot (-144ms). These high inaccuracies are likely due to using $AngVel_{ML}$ signal alone, which may be heavily dependent on the exact orientation of the sensor. Hence, the method may not be as robust, compared to other methods, under field conditions where some degree of sensor movement relative to the horse is inevitable. Indeed, $M1_c$, which uses only resultant acceleration data and is as such more robust to changes in sensor orientation, was more accurate and precise than either, $M3_c$ or $M4_c$. Furthermore, it also outperformed them in terms of sensitivity and PPV, consistently achieving values of 96% and higher compared to 93% for $M3_c$ and 81% for $M4_c$.

The errors incurred by $M3_c$ and $M4_c$ in this study were substantially larger than those reported in the literature. In the source paper, $M3_c$ was reported to incur a bias of 0.6% and 0.1% of a stride cycle for forelimb hoof-on and -off events at trot [24]. In the present application, for hoof-on and -off, the method incurred errors of -26ms and 62ms, respectively, equivalent to -3% and 7% of a stride cycle. These errors are some 5 and 70 times greater than in the original source. $M4_c$ was reported to incur mean errors ranging from -5.4ms to 14.2ms [9], far lower than those observed here (-72ms to 161ms). Several reasons for these substantial differences could be suggested. Firstly, in the present study, a highly varied cohort of horses was recruited and both fore- and hindlimbs tested. In comparison, $M3_c$ had only previously been validated for the forelimbs of trotters at trot and $M4_c$ only for Warmblood horses. The different gait styles adopted by different horses may be partly responsible for the difference in accuracies [19]. Last but not least, different gold standard methods used in the various studies may have also contributed to the difference in errors. Further studies, based on the same gold standard, would be needed to verify this hypothesis. Overall, according to the above considerations, the novel method M1 is recommended when using data collected at the cannon.

**Table 2. Descriptive statistics of errors incurred by each method to detect gait events.**

| (ms) | | | Method | Mean | SD | LLoA | ULoA | ICC | Sensitivity (%) | PPV (%) |
|---|---|---|---|---|---|---|---|---|---|---|
| **FORELIMB** | | | | | | | | | | |
| WALK | Hoof-on | Pastern | **M1** | **-5** | **11** | **-26** | **16** | **0.9981** | **100** | **99** |
| | | | M2 | -4 | 14 | -33 | 24 | 0.9953 | 99 | 99 |
| | | Cannon | M1 | -5 | 21 | -47 | 37 | 0.9905 | 100 | 97 |
| | | | M3 | -30 | 40 | -109 | 49 | 0.9679 | 100 | 93 |
| | | | M4 | 14 | 72 | -127 | 155 | 0.9254 | 90 | 100 |
| | Hoof-off | Pastern | **M1** | **3** | **16** | **-28** | **34** | **0.9957** | **100** | **99** |
| | | | M2 | -7 | 27 | -61 | 46 | 0.9881 | 99 | 99 |
| | | Cannon | **M1** | **1** | **19** | **-36** | **38** | **0.9939** | **100** | **97** |
| | | | M3 | -15 | 65 | -142 | 112 | 0.9287 | 100 | 93 |
| | | | M4 | -35 | 117 | -265 | 194 | 0.8456 | 90 | 100 |
| TROT | Hoof-on | Pastern | M1 | -4*,^ | 10 | -24 | 15 | 0.9912 | 99 | 99 |
| | | | **M2** | **-2*,†** | **9** | **-19** | **15** | **0.9938** | **99** | **99** |
| | | Cannon | M1 | -9^,† | 23 | -55 | 37 | 0.9515 | 99 | 100 |
| | | | M3 | -26 | 21 | -67 | 16 | 0.9567 | 100 | 94 |
| | | | M4 | 43 | 67 | -89 | 175 | 0.8336 | 81 | 100 |
| | Hoof-off | Pastern | M1 | -18* | 23 | -62 | 27 | 0.9581 | 99 | 99 |
| | | | **M2** | **4*,†** | **34** | **-63** | **71** | **0.9373** | **99** | **99** |
| | | Cannon | M1 | -15† | 33 | -80 | 50 | 0.9341 | 99 | 100 |
| | | | M3 | 62 | 114 | -231 | 218 | 0.9559 | 100 | 94 |
| | | | M4 | 77 | 102 | -123 | 277 | 0.7366 | 81 | 100 |
| **HINDLIMB** | | | | | | | | | | |
| WALK | Hoof-on | Pastern | **M1** | **2*,^** | **10** | **-19** | **22** | **0.9974** | **100** | **100** |
| | | | **M2** | **-1*,†** | **10** | **-22** | **19** | **0.9979** | **99** | **100** |
| | | Cannon | M1 | -5^,† | 11 | -26 | 17 | 0.9977 | 100 | 99 |
| | | | M3 | -31 | 55 | -139 | 77 | 0.9456 | 100 | 97 |
| | | | M4 | 161 | 412 | -652 | 974 | 0.7013 | 90 | 100 |
| | Hoof-off | Pastern | **M1** | **6^** | **14** | **-22** | **34** | **0.9934** | **100** | **100** |
| | | | M2 | -3† | 38 | -77 | 72 | 0.9603 | **99** | **100** |
| | | Cannon | **M1** | **-1^,†** | **15** | **-29** | **28** | **0.9940** | 100 | 99 |
| | | | M3 | -57 | 69 | -292 | 77 | 0.9284 | 100 | 97 |
| | | | M4 | -15 | 180 | -368 | 338 | 0.6999 | 90 | 100 |
| TROT | Hoof-on | Pastern | **M1** | **1*,^** | **12** | **-23** | **26** | **0.9947** | **99** | **100** |
| | | | M2 | -1*,† | 19 | -37 | 36 | 0.9880 | 99 | 100 |
| | | Cannon | M1 | -3^,† | 16 | -34 | 29 | 0.9858 | 96 | 100 |
| | | | M3 | -4 | 27 | -56 | 48 | 0.9630 | 100 | 97 |
| | | | M4 | 98 | 147 | -190 | 387 | 0.5439 | 90 | 100 |
| | Hoof-off | Pastern | **M1** | **2*,^** | **9** | **-16** | **20** | **0.9967** | **99** | **100** |
| | | | M2 | 15*,† | 21 | -26 | 57 | 0.9773 | 99 | 100 |
| | | Cannon | M1 | -7^,† | 14 | -35 | 22 | 0.9881 | 96 | 100 |
| | | | M3 | -144 | 62 | -264 | -23 | 0.9128 | 100 | 97 |
| | | | M4 | -72 | 100 | -267 | 124 | 0.6402 | 90 | 100 |

Descriptive statistics of the errors (ms) incurred by each method in detecting gait events and agreement with $M_{ref}$ for calculating stride durations; superscripts *, ^ and † indicate where the difference between two mean errors was statistically significant ($p<0.01$), as revealed by post hoc tests; sensitivity and positive predictive values (PPV) are given as percentages of the total number of events detected; for each gait event, the best performing method(s) is emboldened.

**Table 3. Descriptive statistics of errors incurred by best methods to calculate stance duration on each surface.**

| (%) | Surface | Method | Mean | SD | LLoA | ULoA | Sensitivity (%) | PPV (%) |
|---|---|---|---|---|---|---|---|---|
| **FORELIMB** | | | | | | | | |
| WALK | Asphalt | $M1_p$ | 0.92 | 2.98 | -2.06 | 3.90 | 100 | 99 |
| | Grass | $M1_p$ | 0.52 | 2.14 | -1.61 | 2.66 | 98 | 99 |
| | Sand | $M1_p$ | 1.19 | 5.29 | -4.10 | 6.49 | 100 | 97 |
| TROT | Asphalt | $M2_p$ | 1.09**,*** | 8.68 | -7.59 | 9.77 | 99 | 100 |
| | Grass | $M2_p$ | -0.57** | 6.67 | -7.24 | 6.10 | 100 | 100 |
| | Sand | $M2_p$ | -1.09*** | 7.68 | -8.77 | 6.58 | 99 | 100 |
| **HINDLIMB** | | | | | | | | |
| WALK | Asphalt | $M1_p$ | 0.10 | 3.42 | -3.31 | 3.52 | 100 | 100 |
| | Grass | $M1_p$ | -0.33 | 7.42 | -7.75 | 7.09 | 99 | 100 |
| | Sand | $M1_p$ | 0.22 | 3.60 | -3.38 | 3.82 | 99 | 98 |
| TROT | Asphalt | $M1_p$ | 0.50 | 2.38 | -1.87 | 2.88 | 99 | 99 |
| | Grass | $M1_p$ | 0.92 | 2.26 | -1.34 | 3.18 | 97 | 100 |
| | Sand | $M1_p$ | 1.16 | 4.28 | -3.12 | 5.44 | 98 | 100 |

Descriptive statistics of the errors (as % of stride duration) incurred by the best methods to calculate stance duration; superscripts **p<0.01 and ***p<0.001 indicate where two mean errors differed significantly, as revealed by post hoc tests.

Applications of the novel methods M1 and M2 to the pastern data led to even better results than those found for the cannon. For most events, although the average errors were similar in many cases, the distribution of the error was usually smaller for $M1_p$ than $M1_c$. In fact, $M1_c$ performed better than $M1_p$ in only one case- hindlimb hoof-off at walk- where the SDs were similar but magnitude of the mean error was 5ms bigger for the pastern data than cannon.

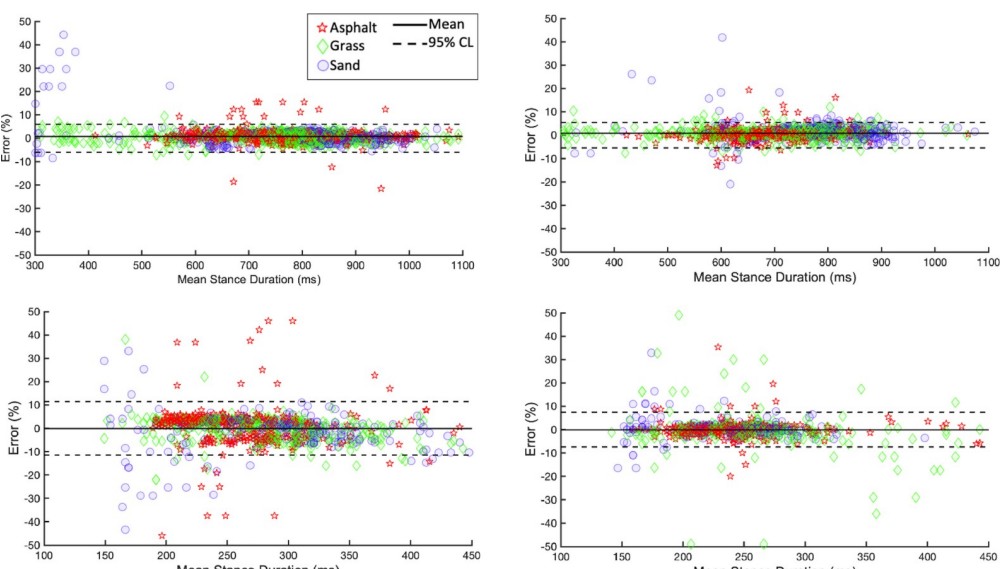

**Fig 6. Bland-Altman plots of novel methods compared to reference method to calculate stance duration on each surface.** The difference between stance durations calculated using reference method and best novel method for each event (shown as a percentage of stride duration) as a function of the mean of the two. Stance durations from control surface are shown as red stars, from grass as green diamonds and from sand as blue circles. The solid horizontal lines (-) indicate the mean error and dashed horizontal lines (- -) the ULoA and LLoA.

This difference was statistically significant but a 5ms (<0.5% of a stride duration) difference is only marginal. In light of this difference being very small and the convenience of having only one site of sensor placement for all events, the pastern was deemed the best location at which to record data for gait event detection.

The higher precision of $M1_p$ is likely due to the pasterns being closer to the site of the hoof impact and lift-off than the cannons, where accelerations associated with the gait event at the hoof might be more attenuated [28] and there is less chance that the peaks of interest could become lost in the noise of the signal. Indeed, it was previously reported that 21% of the initial impact vibration of the forelimb hoof-on remains after the junction of the middle and proximal phalanx, where the pastern sensor is mounted, compared to only 13% after the junction of the proximal phalanx and third metacarpal, where the cannon sensor is mounted [28].

The superiority of acceleration-based methods was confirmed by the comparison between M1 and M2, with the latter performing slightly worse for most events even if still having overall good accuracy (-7 to 15ms) and precision (9 to 38ms). The only cases in which $M2_p$ was more accurate than $M1_p$ was at trot for both forelimb hoof-on (-2ms compared to -4ms) and hoof-off (4ms compared to -18ms).

Considering the results obtained for the event detections, $M2_p$ is recommended to calculate forelimb stance durations at trot and $M1_p$ for all other cases.

It has previously been reported that mild and moderate forelimb lameness can cause reductions in trot stride duration of 11ms and 31ms, respectively [8]. The mean errors incurred by the described pastern-based methods to detect hoof-on events at trot (-2ms for forelimbs and 1ms for hindlimbs) are small enough that the methods would be reliable for measuring such changes.

## 4.2 Comparison between surfaces

When using $M1_p$ and $M2_p$ as recommended, stance durations could be calculated with very high accuracy, with mean errors <1.5% of a stride cycle for all limbs and gaits on all surfaces. The errors on asphalt were lower than those reported in the literature on an equivalently hard surface (laboratory), which ranged from -0.8% of a stride duration for the hindlimbs at walk to 9.1% for the hindlimbs at trot [9].

In the literature, it was reported that unilateral forelimb lameness increased the stance phase of the ipsilateral hindlimb by 1.3% and that of both forelimbs by 2.3% in horses trotting on a treadmill [7]. The mean error in trotting stance durations calculated using the described methods range from 0.1 to 1.16% of a stride duration, suggesting that the methods are sufficiently accurate to detect these lameness-dependent changes.

Small differences between the surfaces were observed for the mean errors and SD of the errors, but none of these differences were of noteworthy magnitude, and they were only statistically significant for the forelimbs at trot. Therefore, it is concluded that the chosen methods had an equivalent performance on all surfaces when used to calculate stance durations.

In the case of the hindlimbs at trot, the Bland-Altman plots appear to show that the method tended to underestimate the stance duration for mean stance durations of over 300ms. However, the points above 300ms which fell below the LLoA (10 points) are very few compared to the points which fell within the LoA. Conversely, the Bland-Altman plot for the hindlimbs at trot suggest that the method tended more frequently to overestimate the stance duration for mean stance durations of below 300ms. However, the points here falling above the ULoA, compared to those falling within the LoA, are again only a small minority. These two observations could warrant further testing to determine whether there is a trend between the duration of the stance phase and whether the methods tend to more frequently under- or overestimate the stance duration.

No significant effects due to the individual horse were seen and the Bland-Altman plots suggest that the errors incurred were consistent for different durations of stance for both limbs and gaits. This indicates that the method can be used for different horses and also in different surface conditions.

### 4.3 Limitations

The most significant limitations to this research relate to the use of the hoof-based method [13] as reference values. This method has only been validated for data collected in the lab, on one hard surface, with results compared to a force plate. However, a previous study which used a hoof-mounted accelerometer to record accelerations [3], found that gait events could be manually selected from signals recorded on both hard and soft surfaces. Furthermore, visual inspection of $Acc_R$ and $AngVel_R$ recorded from the hooves on the softer surfaces in the present study, revealed that they were comparable to those recorded on asphalt, with similar peaks prominent. Therefore, it was assumed that the hoof-based method held for all surfaces. Although a varied group was selected in the interests of yielding widely applicable results, the cohort size, while larger than in many similar studies, was still somewhat limited and it would be beneficial to validate the methods for more individuals. In the future, use of pastern mounted sensors should also be investigated for other gaits.

## 5 Conclusions

In conclusion, this paper has compared gait event detection using different processing methods applied to data collected by IMUs attached to the fore- and hindlimb pasterns and cannons. The performance of these methods was consistent across the entire, varied cohort. Pastern-based methods have proven to be superior to the current state of the art cannon-based alternatives. Stance durations calculated using events detected by a method of peak detection applied to the resultant angular velocity (for forelimbs at trot) and resultant linear acceleration (for all other events) recorded at the pastern level, incurred consistent errors for data recorded on asphalt, grass and sand. The methods developed here enable gait event detection under a range of surface conditions and for a varied cohort of subjects. This can certainly be extremely beneficial for future studies undertaken in a variety of field conditions.

## Acknowledgments

The authors wish to thank Marij Tijssen, for her assistance in implementing her published methods of gait event detection and Dr Filipe Serra Bragança for advice regarding application of his methods. Further thanks go to Dr Caroline Clarke for the loan of her horses and her assistance in data collection.

## Author Contributions

**Conceptualization:** Eloise V. Briggs, Claudia Mazzà.

**Formal analysis:** Eloise V. Briggs.

**Investigation:** Eloise V. Briggs.

**Writing – review & editing:** Eloise V. Briggs, Claudia Mazzà.

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
