## [Decision Letter · Decision Letter 0]

11 Jun 2021

PONE-D-21-07648

Detection of equine gait events during walk and trot on asphalt, grass and sand using wearable inertial sensors

PLOS ONE

Dear Dr. Briggs,

Thank you for submitting your manuscript to PLOS ONE. After careful consideration, we feel that it has merit but does not fully meet PLOS ONE’s publication criteria as it currently stands. Therefore, we invite you to submit a revised version of the manuscript that addresses the points raised during the review process.

My apologies for the delay in obtaining sufficient reviewers for the manuscript. Please have a look at the useful comments of reviewer two when revising the manuscript.

We look forward to receiving your revised manuscript.

Kind regards,

Chris Rogers

Academic Editor

PLOS ONE

Journal Requirements:

"The research of EVB is funded by Worldbase Ltd. and this study was also supported by the UK EPSRC (EP/K03877X/1, EP/S032940/1, https://epsrc.ukri.org). The funders had no role in study design, data collection and analysis, decision to publish, or preparation of the manuscript."

We note that you received funding from a commercial source: Worldbase Ltd.

Reviewers' comments:

Reviewer's Responses to Questions

**Comments to the Author**

1. Is the manuscript technically sound, and do the data support the conclusions?

Reviewer #1: Partly

Reviewer #2: Yes

2. Has the statistical analysis been performed appropriately and rigorously? 

Reviewer #1: Yes

Reviewer #2: Yes

3. Have the authors made all data underlying the findings in their manuscript fully available?

Reviewer #1: Yes

Reviewer #2: Yes

4. Is the manuscript presented in an intelligible fashion and written in standard English?

Reviewer #1: Yes

Reviewer #2: Yes

5. Review Comments to the Author

Reviewer #1: This paper presents two new algorithm to detect accurately hoof on and off event. The paper is in overall well written but there is a lack of accurate definition of what type of event they are looking at. In fond the title misleading since I was expecting to discover an algorithm which allows the differentiation between walk and trot, while it only detects, within a gait, when the foot is on or off. They sometimes use values in percentage and sometimes in ms to compare the different methods without really explaining why they use those two different units rather than one. Moreover some of the statistical results summary are confusing because not using the common standards.

Once all of this would have been modified, this paper can bring two interesting algorithms in the field of horse motion detection.

Reviewer #2: This article is generally well-written and provides a useful addition to the current literature around automated gait event detection in horses, especially with the inclusion of different surfaces and breed types.

I am curious as to why the authors relied on event detection rather than using machine learning or neural networks, as machine learning is becoming more common in this field. The accuracy reported here is acceptable, but it would be interesting to see if increased accuracy could be gained via different techniques. A brief note somewhere on the decision to use the selected algorithms rather than a machine learning approach would be useful.

The introduction would benefit from more detail on the potential usefulness of gait event detection (e.g., equitation related research, lameness diagnosis etc.) for those not familiar with the field. Overall I would like to see more in depth assessment of the existing literature in both the introduction and the discussion.

Regarding the use of force plates, I agree that they are impractical under most circumstances– but there have been limited studies using force plates in the field (e.g., Self Davies et al. 2019) but the complexity of embedding force plates under tracks is an obvious disadvantage.

In the discussion and/or limitations, some attention should be paid to how acceptable the reported errors are from a biological standpoint. The errors are small, but depending on the intended use of this technology, the errors may limit the usefulness of the sensors. The sample frequency may also be a limitation here if there is any intention to validate for faster gaits later on- 200Hz is likely limiting for detecting subtle variations at the gallop.

Specific comments

Line 55-58 – Consider re-wording, as the use of the pastern sensor and the fact the methods are not available aren’t quite clear

Line 73 – Need to specify that the figures in brackets (presumably) indicate standard deviation

Line 82 – Ideally use anatomical terms the first time the locations of sensors are specified (e.g., pasterns (proximal phalanges), cannon bones (third metacarpal bones)) as lay nomenclature may vary globally.

Line 88 – fore- and hindlimbs

Line 91- Were the surfaces level? If the information is available, it would also be helpful to specify how firm the grass surface was as there can be considerable variation which would likely affect the accuracy of the sensors. It would also be helpful to specify if there was there any variation in the grass surface between trials, or if the trials were all performed on the same day.

Line 101 – please double check manufacturer citation, I think this should be The MathWorks not MathWorks.

Line 106- fore- and hindlimb (please check throughout)

Line 108 – Add a citation here to make it clear if you are still referring to reference 4

Line 155 & after – consider trough instead of valley?

Line 202 – why was p>0.01 used rather than p<0.05?

Lines 230 – 232 – consider adding ‘respectively’ to sentence for clarity

Line 254 – Specify if figure in brackets is SD the first time you use it in this section

Line 314 – I agree that some movement relative to the horse is inevitable. Have you or others quantified the amount of sensor movement when attached at the cannon/pastern? If so please cite/specify this.

Lind 377-378 – The hoof mounted technique has been used elsewhere at a variety of gaits – check Witte et al. 2004

Line 400 – typo for ‘event’

References

Self Davies, Z.T., Spence, A.J., Wilson, A.M. (2019) ‘Ground reaction forces of overground galloping in ridden Thoroughbred racehorses’, Journal of Experimental Biology, 222(16).

Witte, T.H., Knill, K., Wilson, A.M. (2004) ‘Determination of peak vertical ground reaction force from duty factor in the horse (Equus caballus)’, Journal of Experimental Biology, 207(21), 3639–3648, available: http://jeb.biologists.org/cgi/doi/10.1242/jeb.01182.

6. PLOS authors have the option to publish the peer review history of their article (what does this mean?). If published, this will include your full peer review and any attached files.

Reviewer #1: **Yes: **Amandine Schmutz

Reviewer #2: No

---

## [Author Response · Author response to Decision Letter 0]

28 Jun 2021

The authors would like to extend their thanks to the Editor and Reviewers for their insightful comments and suggestions. Revisions and amendments of the manuscript have been made, accordingly, and we believe these have strengthened the article.

Specific responses to the Editor and Reviewers can be found in the accompanying document- Response to Reviewers.

---

## [Editor Report · Decision Letter 1]

5 Jul 2021

Automatic methods of hoof-on and -off detection in horses using wearable inertial sensors during walk and trot on asphalt, sand and grass

PONE-D-21-07648R1

Dear Dr. Briggs,

We’re pleased to inform you that your manuscript has been judged scientifically suitable for publication and will be formally accepted for publication once it meets all outstanding technical requirements.

Kind regards,

Chris Rogers

Academic Editor

PLOS ONE

Additional Editor Comments (optional):

Thank you for the edits to the manuscript.
---

## [Editor Report · Acceptance letter]

14 Jul 2021

PONE-D-21-07648R1 

Automatic methods of hoof-on and -off detection in horses using wearable inertial sensors during walk and trot on asphalt, sand and grass 

Dear Dr. Briggs:

I'm pleased to inform you that your manuscript has been deemed suitable for publication in PLOS ONE. Congratulations! Your manuscript is now with our production department. 

Kind regards, 

on behalf of

Dr. Chris Rogers 

Academic Editor

PLOS ONE